# Efficacy and safety of traditional Chinese medicine nasal irrigation on chronic rhinosinusitis recovery after endoscopic sinus surgery: A protocol for a systematic review and meta-analysis

**Yepeng Yang**[1,2], **Yaning Sun**[1,2], **Feng Xiang**[1,2], **Min Zhang**[1,2], **Li Fu**[1,2], **Qinxiu Zhang**[1,2,3]*

**1** Hospital of Chengdu University of Traditional Chinese Medicine, Chengdu, Sichuan Province, P.R. China, **2** College of Clinical Medicine, Chengdu University of Traditional Chinese Medicine, Chengdu, Sichuan Province, P.R. China, **3** World Health Organization Collaborating Centre, CHN-56, Chengdu, Sichuan Province, P.R. China

* zhangqinxiu@cdutcm.edu.cn

**Data Availability Statement:** Deidentified research data will be made publicly available when the study is completed and published.

## Abstract

### Background

Continuous comprehensive treatment is still needed after endoscopic sinus surgery (ESS) for chronic rhinosinusitis (CRS) to promote the recovery of sinus mucosal morphology and function. Traditional Chinese medicine (TCM) nasal irrigation is a promising external treatment of TCM, but at present, the application of TCM nasal irrigation after ESS for CRS has not been recommended by the guidelines. Therefore, this article aims to develop a systematic overview and meta-analysis protocol to assess the effectiveness and safety of Chinese herbal nasal rinse for CRS recovery after ESS.

### Methods

Seven databases shall be retrieved from their inception until December 2021. Eligible randomized controlled trials will be covered in the study. The outcome indicators of the survey will consist of efficacy, visual analogue scale score, Lund-Kennedy score for nasal endoscopy, Lund-Mackay score for sinus computed tomography and other secondary outcome indicators. The selection of literature, extraction of data, and methodological quality evaluation of literature shall be conducted by two researchers separately. If there is any dispute, it can be discussed and solved by a third researcher. Review Manager 5.3 software will be applied to data analysis.

### Results

The article will make a detailed research programme to explore the efficacy and safety of TCM nasal irrigation on CRS recovery after ESS.

**Funding:** This work was supported by a Grant from Xinglin Scholars Scientific Research Promotion Plan of Chengdu University of Traditional Chinese Medicine-Innovation team of traditional Chinese medicine otorhinolaryngology discipline, natural science(XKTD2021003). URL:https://www.cdutcm.edu.cn/. The funders had and will not have a role in study design, data collection and analysis, decision to publish, or preparation of the manuscript.

**Competing interests:** The authors have declared that no competing interests exist.

**Abbreviations:** CRS, chronic rhinosinusitis; ESS, endoscopic sinus surgery; TCM, traditional Chinese medicine; HPAA, hypothalamic-pituitary-adrenal axis; PRISMA-P, Preferred Reporting Items for Systematic Reviews and Meta-analysis Protocol; Embase, Excerpt Medical Database; CENTRAL, Cochrane Central Register of Controlled Trials; CBM, Chinese Biomedical Literature Database; CNKI, China National Knowledge Infrastructure; VIP, Chinese Science and Technology Periodical Database; RR, relative risk; CI, onfidence interval; MD, mean differences; SMD, standardized mean difference.

## Conclusion

This protocol is suitable for evaluating the effectiveness and safety of TCM nasal rinse for CRS recovery after ESS, and can provide corresponding evidence-based medical evidence.

## Systematic review registration

Open Science Framework Registration DOI: 10.17605/OSF.IO/ZV73Q.

## Introduction

Chronic rhinosinusitis (CRS) is a common disease in otorhinolaryngology, impacting 5–12% of the general population [1]. The overall prevalence of CRS in the Chinese population is 8% [2,3], slightly lower than 10.9% in Europe [4] and 12% in the United States [5]. CRS is a highly heterogeneous disease, and its pathogenesis is related to the chronic obstruction of the ostio-meatal complex caused by anatomical differences, viral and bacterial infections, allergies, genetics and other factors [6]. The Chinese epidemiological survey data showed that 11.2% of CRS patients had asthma, and 27.3% had airway hyperresponsiveness [7]. A systematic review conducted by Antonino et al. showed that CRS was closely associated with atopy (49.9%), asthma (50.33%), samter triad (4.9%), and eosinophilia (4.28%) [8]. CRS greatly affects the social economy and patients' quality of life. In the United States, the total direct costs associ-ated with CRS range from $10 billion to $13 billion each year; the total overhead cost of job productivity losses associated with CRS is estimated to exceed $20 billion per year [9]. In addi-tion, persistent symptoms can also affect the patient's life and work, and even lead to problems such as depression or anxiety.

At present, there are two main treatment methods for chronic sinusitis: drug therapy and surgery. After ineffective drug treatment, endoscopic sinus surgery (ESS) is a standard surgical treatment method. The endoscopic treatment represents the best therapeutic option that the surgeon can offer for invasiveness and safety, allowing quick post-surgical, less postoperative pain and fewer complications [10]. However, surgery cannot change the inflammatory nature of the sinus mucosa, and continuous surgical cavity nursing and comprehensive drug therapy are still required to promote the gradual recovery of the morphology and function of the sinus mucosa [11]. In comprehensive postoperative treatment, seeking more effective and ideal treatment measures to stimulate the healing of the mucosal morphology and function of the surgical cavity is an important research topic faced by contemporary rhinologists.

As an essential adjuvant therapy after ESS, nasal irrigation has achieved specific curative effects in clinical application and has the advantage of fewer side effects. In addition to mechanical drainage to clean the postoperative nasal cavity, nasal irrigation can also reduce sinus mucosal inflammation, improve mucociliary clearance, and accelerate the recovery of sinus mucosal structure and function [12–15]. At present, postoperative nasal irrigation solu-tions include normal saline, hypertonic saline, normal saline with corticosteroids and (or) antibiotics, Chinese medicine liquid, etc. Normal saline causes no irritation to the nasal mucosa and is the most commonly used nasal rinse after surgery, but its curative effect is lim-ited. Hypertonic saline can reduce nasal mucosal edema, but a high concentration of saline will inhibit the motor function of cilia. Jiao et al. found that hypertonic 3.0% saline signifi-cantly reduced the epithelial mucociliary or barrier function of cultured human nasal epithelial

cells. Moreover, it had apparent cytotoxic effects on human nasal epithelial cells [16]. Gluco-corticoids have a significant effect on reducing mucosal edema and polyp formation in the surgical cavity and promoting mucosal recovery [17]. A meta-analysis of studies by Yoon et al. showed that the beneficial effect of additional steroids in saline irrigation compared with saline irrigation alone was equivocal in terms of endoscopic score and CRS-related quality of life improvement [18]. Soudry et al. revealed that long-term nasal rinse with budesonide is generally safe, but some patients may experience asymptomatic hypothalamic-pituitary-adrenal axis (HPAA) suppression. Simultaneous use of a nasal steroid spray and a pulmonary steroid inhaler increases the risk of HPAA, especially with daily budesonide nasal irrigation [19].

Over the recent years, traditional Chinese medicine (TCM) nasal irrigation has gradually attracted the attention of researchers. More and more clinical trials have evaluated the effect of TCM nasal rinse on CRS recovery after ESS, indicating that it may be an effective and safe treatment method. Studies have revealed that [20–24] local nasal irrigation with TCM is effective in reducing nasal mucosa edema, promoting epithelialization of surgical cavity mucosa, and promoting the recovery of nasal mucociliary structure and motor function. Moreover, the local action mechanism of TCM is not a single antibacterial and anti-inflammatory, but also related to local immune regulation. Chinese medicine nasal irrigation has high safety and no obvious adverse reactions. Before and after treatment, the patient's liver and kidney functions were normal. Only a few patients experienced nasal pain, nasal itching, and headache during nasal irrigation [24–26]. A systematic review published in 2011 showed the effectiveness and safety of TCM nasal irrigation for the postoperative recovery of chronic sinusitis after ESS [27]. However, many related clinical trials and new efficacy evaluation indicators have been added in recent years. Therefore, this paper aims to formulate a research plan to explore whether Chinese herbal nasal irrigation is effective and safe for the recovery of chronic sinusitis after ESS.

## Methods

### Protocol registration

The research protocol has been registered on the Open Science Framework (OSF) under the registration DOI 10.17605/OSF.IO/ZV73Q (http://osf.io/zt7h9).

According to the Preferred Reporting Items for Systematic Reviews and Meta-analysis Protocol (PRISMA-P) [28], We accomplished this research protocol. This research is a secondary acquisition and analysis of the data, and therefore no ethical approval or patient informed consent is required.

### Inclusion and exclusion criteria

**Types of research.** All randomized controlled trials evaluating the effectiveness and safety of TCM nasal irrigation for CRS recovery after ESS shall be contained. The deadline for the included documents is December 31,2021. The postoperative nasal irrigation time of patients should be at least one month. If it is less than one month, the literature will not be included. Languages are confined to English and Chinese, with no restrictions on publication type and blindness. Duplicate publications, non-randomized controls, narrative reviews, case reports, animal tests, and other unrelated papers will be excluded.

**Participants.** All cases must meet the diagnostic criteria of this disease [11] and have undergone nasal endoscopic sinus surgery. After the operation, nasal irrigation was performed on the basis of standard treatment, regardless of sex, age, and race. Trials with allergic rhinitis shall be eliminated.

**Interventions.** The experimental group is irrigated with Chinese medicine, and the control group is irrigated with normal saline alone. The experimental group is given Chinese medicine combined with normal saline nasal rinse, while the control group is given normal saline plus antibiotics and (or) glucocorticoid nasal rinse. The experimental group is treated with Chinese medicine combined with normal saline plus antibiotics and (or) glucocorticoid nasal rinse, while the control group is treated with normal saline plus antibiotics and (or) glucocorticoid nasal irrigation.

**Outcome indicators.** Main outcome indicators: efficacy, including effective rate and cure rate, visual analogue scale score, Lund-Kennedy score of nose endoscope, and Lund-Mackay score of nasal sinus computed tomography.

Secondary outcome indicators: sino-nasal outcome test-20 scale, sino-nasal outcome test-22 scale, medical outcome study short form 36-items health survey, mucociliary transit time or mucociliary transport rate, cleaning time of surgical cavity, epithelization time of operative cavity, nasal immune function index determination, adverse reactions.

## Retrieval strategies

These seven databases shall be retrieved from their inception to December 2021: PubMed, Excerpt Medical Database (Embase), Cochrane Central Register of Controlled Trials (CENTRAL), Chinese Biomedical Literature Database (CBM), China National Knowledge Infrastructure (CNKI), Chinese Science and Technology Periodical Database (VIP) and Wan Fang Database. Retrieval strategies of the PubMed database are displayed in Table 1, and the search strategies of other databases are adjusted accordingly. In addition, ongoing clinical trials can be searched in the Chinese clinical trial registry. References to all relevant papers can be searched manually to facilitate access to more experimental and related research data. We can get in touch with the author by email for more details, if necessary.

## Selection of literature

According to the above search strategy, firstly, two independent reviewers (Yang YP and Sun YN) will conduct a literature search, and obtained literature will be imported into Endnotes X9 for deduplication. Then, two independent reviewers (Yang YP and Sun YN) screen through reading headlines and summaries, download the full text of the document that initially meets the standards, and decide on the inclusion and exclusion of the document after reading the full text. If they disagree, they can discuss with the third reviewer (Zhang QX) to decide whether to include the study. The selection process of literature is based on the PRISMA flow diagram (Fig 1).

**Table 1. Search strategy for PubMed.**

| Number | Search terms |
|---|---|
| #1 | (chronic sinusitis) OR (chronic rhinosinusitis) OR (sinusitis) OR (rhinosinusitis) OR (nasosinusitis) OR (nasal sinusitis) |
| #2 | (endoscopic sinus surgery) OR (functional endoscopic sinus surgery) OR (endoscopic surgery) OR (nasal endoscope) OR (nose endoscope) |
| #3 | (nasal irrigation) OR (nasal cavity irrigation) OR (nasal lavage) OR (nasal douche) OR (nasal rinse) OR (nasal wash) |
| #4 | (traditional Chinese medicine) OR (Chinese medicine) OR (Chinese herb) |
| #5 | #1 AND #2 AND #3 AND #4 |
| #6 | (randomized controlled trial) OR (clinical study) OR (clinical trial) OR (controlled clinical trial) |
| #7 | #5 AND #6 |

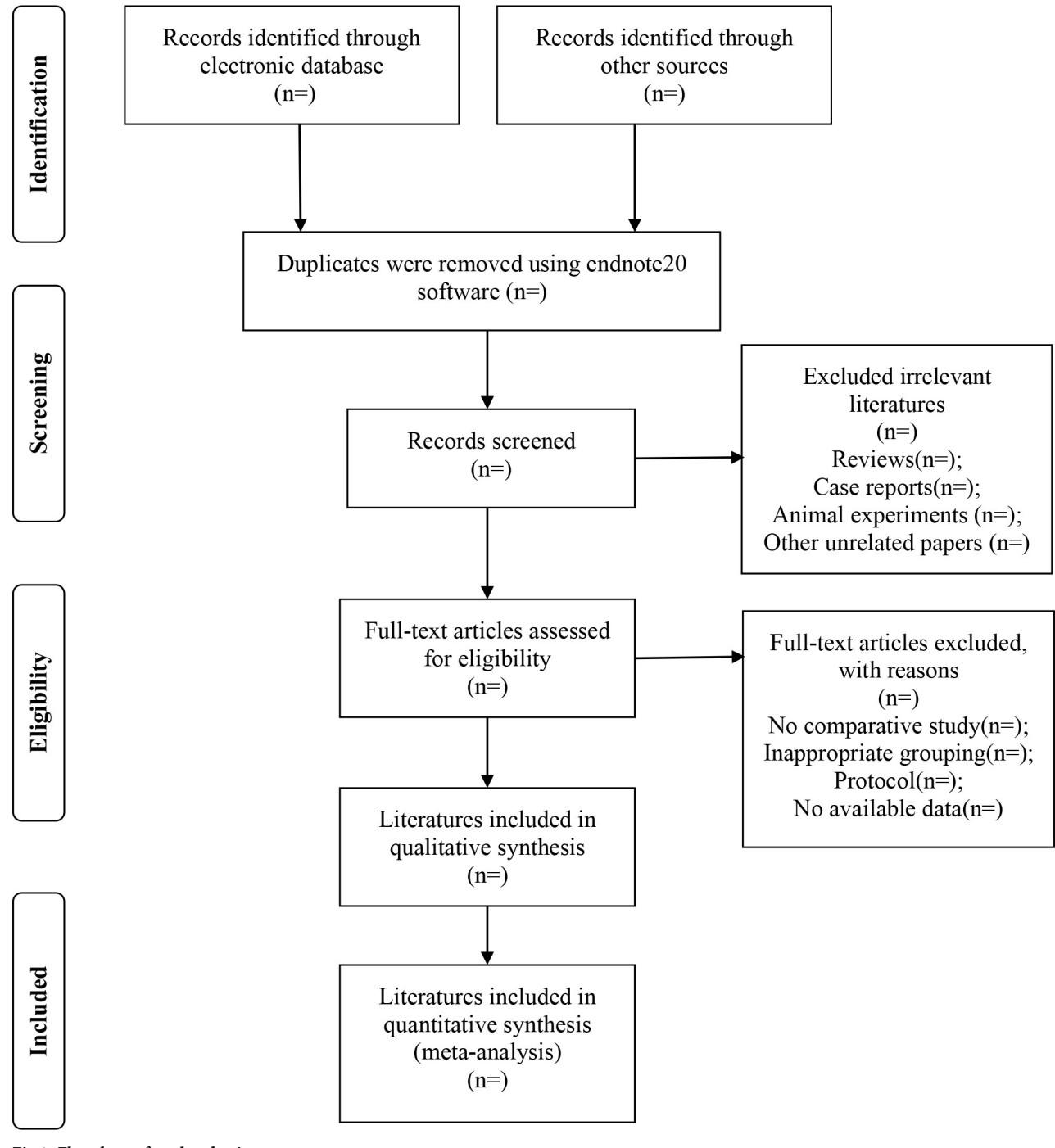

**Fig 1. Flowchart of study selection.**

## Data extraction

Two independent investigators (Xiang F and Zhang M) will respectively extract the publication year, author's name, sample size, patient characteristics (such as age, course of the disease, etc.), detailed information of intervention and control measures (such as the specific composition and dosage of TCM), outcome indicators and literature quality (such as randomized

control method, blinding, allocation concealment, etc.), adverse reactions and other relevant information. For incomplete data, we shall contact the article's authors to supplement associated data. For different opinions, we discuss with Zhang QX to resolve the dispute.

## Methodological quality assessment

Xiang F and Fu L will evaluate the relevant literature methodology on the basis of the bias risk assessment tool in the Cochrane Handbook for Systematic Reviews of Interventions (version 5.1.0) [29], and apply RevMan 5.3 software to produce an example graph of risk bias assessment results. If there is any disagreement, we will discuss the decision with Zhang Qinxiu.

Here are seven bias items: random sequence generation (selection bias), allocation concealment (selection bias), blinding of subjects and researchers (implementation bias), blinding of outcome assessors (measurement bias), incomplete outcome data (follow-up bias), selective reporting (reporting bias), other bias. Each item will be judged as "low risk of bias", "high risk of bias" and "unclear" according to the risk of bias assessment criteria.

## Data analysis

Review Manager 5.3 will be applied to data analysis. The risk ratio (RR) and 95% confidence interval (95%CI) will be used for the dichotomous data. For continuous measurement data, mean differences (MD) and 95% CI will be used. If the outcome measures of the study are different, the standardized mean difference (SMD) shall be adopted.

Heterogeneity will be assessed using the chi-square test and $I^2$ statistic. If $P > 0.1$ or $I^2 <$ 50% indicates that there is no apparent heterogeneity, a fixed-effects model will be selected; or else, there is significant heterogeneity, and a random-effects model can be applied. For sources of heterogeneity, we can perform subgroup analyses and sensitivity analyses. According to factors such as age, course of the disease, the dose of TCM, and quality of literature, the literature can be grouped, and the literature corresponding to relevant factors can be combined for subgroup analysis to evaluate the impact on outcome indicators. Alternatively, we can eliminate the literature one by one to see if the heterogeneity has changed. If the heterogeneity changes after excluding specific literature, this literature may be the origin of heterogeneity, which may be analyzed from experimental design, sample volume, outcome indicators, etc. If the heterogeneity does not change significantly after each article is eliminated, the results are more reliable. If the data cannot be analyzed quantitatively, we shall make a qualitative description.

If more than 10 studies are included, a funnel chart will be made to determine whether there is publication bias. Finally, the strength of the evidence can be rated using the grades of Recommendations Assessment, Development and Evaluation profiler.

## Discussion

CRS refers to chronic inflammation of the sinus mucosa caused by the dysfunction between all kinds of environmental aspects and the host immune system. In healthy humans, the mucosa is a barrier that modulates interactions with the host immune system, promotes tolerance and symbiosis, and prevents or limits inflammation [1]. In CRS patients, this barrier is penetrated, leading to chronic inflammation [1]. The mucociliary clearance function, humidification and filtration function of inflammatory nasal mucosa will reduce. Patients may develop inferior turbinate hypertrophy and nasal congestion and do not respond to standard medical therapy [30]. Endoscopic sinus surgery can solve the problem of sinus ostium obstruction and drainage, and create the necessary conditions for the benign outcome of sinus mucosal inflammation. However, after ESS, the operative cavity is exposed, secretions are retained, and crusts are accumulated, and some new lesions may occur, such as mucosal edema, vesicle formation, and

polyp regeneration. The nasal mucosa transformation after ESS includes the following four stages: cleaning of the surgical cavity, mucosal transition, complete epithelialization, and tissue remodeling [31]. Therefore, promoting the benign outcome and epithelialization of the mucosa of the surgical cavity and reducing mucosal edema, vesicles, granulation, and other lesions during the postoperative outcome stage is an essential link to treating chronic sinusitis.

Chinese medicine nasal irrigation is a kind of external treatment of TCM, which has the unique advantages of TCM prevention and treatment. While cleaning the nasal cavity, it can also be absorbed and administered through local mucosa to achieve local and systemic therapeutic effects. Chronic sinusitis belongs to the category of "biyuan" in TCM. Chinese medicine believes that the primary pathogenesis of biyuan is wind-heat in the lung meridian, stagnant-heat in the gallbladder, and damp-heat in the spleen and stomach. Therefore, at present, TCM washing liquids are mainly selected with Chinese medicines with the functions of clearing heat and promoting diuresis, purging intense heat and detoxicating, activating circulation and removing blood stasis, and clearing the nasal cavity. Their curative effects have also been well reflected in clinical practice. Sihuang cangtao decoction is composed of coptis, cortex phellodendri, radix scutellariae, cocklebur fruit, peach seed, etc. It has the effects of clearing away heat and expelling pus, tonifying spleen and excreting dampness, activating blood to remove stasis, and dispersing and relieving nose orifice. Sihuang cangtao decoction nasal irrigation can control the inflammatory exudation and scab production of surgical wounds in patients with chronic sinusitis after the operation, prevent granulation tissue hyperplasia and vesicle formation, and improve the symptoms and signs scores of patients after the operation without noticeable adverse reactions [32]. The drug composition of Tongqiao flushing formula includes radix astragali, peach seed, flos magnoliae, radix angelicae dahuricae, platycodon grandiflorum, spina gleditsiae, flos chrysanthemi indici, etc., which has the functions of activating blood to remove stasis, draining the pus and clearing the orifices. Tongqiao irrigation formula used for nasal irrigation after ESS for chronic sinusitis can significantly shorten the epithelization time of the operation cavity, accelerate the recovery of the operation cavity mucosa and improve the postoperative symptoms and signs [33]. In addition, some modern pharmacological studies have proved that flos magnoliae, cocklebur fruit, radix angelicae dahuricae, radix scutellariae, coptis, cortex phellodendri, peach seed and flos chrysanthemi indici have anti-inflammatory effects [34–41]. Coptis polysaccharide can enhance the phagocytosis of macrophages, promote the differentiation, development and maturation of T cells, and play an immunomodulatory role [42]. Radix astragali and peach seed have a two-way regulating effect on the immune system. Aiming at the situation of low immunity, it can improve the immune function of the body. For the inflammatory response caused by hyperimmunity, it can inhibit the inflammatory response of the body [43,44].

Nasal irrigation is an effective auxiliary means for the treatment of chronic sinusitis after ESS. It is beneficial to postoperative nursing and nasal mucosal repair. In 2015, "AAO-HNS Clinical Practice Guidelines: Adult Sinusitis " pointed out that clinicians recommend nasal saline irrigation as an adjuvant treatment for chronic sinusitis, which can moisturize nasal mucosa, remove blood clots and scabs and promote mucosal healing after sinusitis surgery [45]. However, the relevant guidelines do not give opinions on TCM nasal irrigation, which may be caused by the deficiency of evidence-based medical evidence. Thus, this research aims to formulate a study protocol on the effectiveness and safety of TCM nasal irrigation on the recovery after ESS for chronic sinusitis, in order to offer more evidence-based medical support for the use of TCM nasal rinse after ESS for chronic sinusitis.

## Supporting information

**S1 Checklist. PRISMA-P 2015 checklist.**
(DOC)

## Author Contributions

**Conceptualization:** Yepeng Yang, Yaning Sun, Qinxiu Zhang.

**Data curation:** Yepeng Yang, Yaning Sun, Feng Xiang, Min Zhang.

**Formal analysis:** Feng Xiang, Min Zhang.

**Investigation:** Yepeng Yang, Yaning Sun, Li Fu.

**Methodology:** Feng Xiang, Li Fu.

**Project administration:** Yepeng Yang, Qinxiu Zhang.

**Supervision:** Qinxiu Zhang.

**Validation:** Yaning Sun, Qinxiu Zhang.

**Writing – original draft:** Yepeng Yang.

**Writing – review & editing:** Yepeng Yang, Qinxiu Zhang.

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
