## [Decision Letter · Decision Letter 0]

4 May 2022

PONE-D-22-04435Efficacy and safety of traditional Chinese medicine nasal irrigation on chronic rhinosinusitis recovery after endoscopic sinus surgery：A protocol for a systematic review and meta-analysisPLOS ONE

Dear Dr. Zhang,

Thank you for submitting your manuscript to PLOS ONE. After careful consideration, we feel that it has merit but does not fully meet PLOS ONE’s publication criteria as it currently stands. Therefore, we invite you to submit a revised version of the manuscript that addresses the points raised during the review process.

We look forward to receiving your revised manuscript.

Kind regards,

Giannicola Iannella, M.D

Academic Editor

PLOS ONE

Journal Requirements:

Additional Editor Comments:

please revise the manuscript according to the comments of the reviewers

Reviewers' comments:

Reviewer's Responses to Questions

**Comments to the Author**

1. Does the manuscript provide a valid rationale for the proposed study, with clearly identified and justified research questions?

Reviewer #1: Yes

Reviewer #2: Yes

2. Is the protocol technically sound and planned in a manner that will lead to a meaningful outcome and allow testing the stated hypotheses?

Reviewer #1: Yes

Reviewer #2: Partly

3. Is the methodology feasible and described in sufficient detail to allow the work to be replicable?

Reviewer #1: Yes

Reviewer #2: Yes

4. Have the authors described where all data underlying the findings will be made available when the study is complete?

Reviewer #1: Yes

Reviewer #2: No

5. Is the manuscript presented in an intelligible fashion and written in standard English?

Reviewer #1: Yes

Reviewer #2: Yes

6. Review Comments to the Author

You may also provide optional suggestions and comments to authors that they might find helpful in planning their study.

Reviewer #1: The paper is interesting minor corrections could be improve the overall quality:

English editing should be performed.

I should change the titile in PLOS ONE

Efficacy and safety of traditional Chinese'' Effectiveness of traditiona..a systematic review..

Abstract

we used..not We will use Review Manager 5.3 software

Introduction

- Pediatric chronic rhinosinusitis (CRS) is a common disorder that carries significant morbidity. The diagnosis requires sinus symptoms that persist despite standard medical therapy greater than 3 months. Viral infections, allergies, and anatomic differences in children lead to chronic obstruction of the osteomeatal complex. The diagnosis is a conglomeration of multiple phenotypes and endotypes. As such, the diagnosis and management are complex. The emerging potential treatment options of CRS, including anti-immunoglobulin E, interleukin-5, and interleukin-4 receptor alpha subunit. please discuss and cite doi:10.1007/s11882-018-0792-8

- line 63, An interesting systematic review on single-nucleotide polymorphisms and risk-related chronic rhinosinusitis reporting the gene variation implicated in the pathogenesis of chronic inflammation and polyps. 12 papers with 9127 patients, of which 2739 CRS cases and 6388 controls were found. The major comorbidities reported related to chronic rhinosinusitis were atopy in 4555 (49.9%), asthma in 4594 (50.33%), Samter Triad in 448 (4.9%) and eosinophilia in 391 subjects (4.28%). please cite doi:10.1111/coa.13870

- line 67, Endoscopic treatment represents the best therapeutic option that the surgeon can offer for invasiveness and safety, allowing quick post-surgical recovery, less postoperative pain and fewer complications. please discuss and cite doi:10.1007/s00405-021-06724-6

- line 72, Nasal saline irrigation (NSI) plays an important role in the treatment of chronic rhinosinusitis (CRS). It is a beneficial low-risk treatment that serves an adjunctive function in the medical and surgical management of CRS. NSI is hypothesized to function by thinning mucous, improving mucociliary clearance, decreasing edema, and reducing antigen load in the nasal and sinus cavities. Although its use in CRS is nearly universal, significant variety exists with regard to delivery volume, delivery pressure, frequency of use, duration of use, composition, and hygiene recommendations. Evidence is limited regarding the most optimal methods of NSI delivery. In addition, use of NSI has recently come under increasing scrutiny due to potential associations with cases of primary amebic meningoencephalitis. An interestin review provided a clinical update summarizing use of NSI for treatment of CRS, including current recommendations for use, and data regarding overall efficacy, available delivery devices, solution composition, and hygiene. please discuss and cite doi:10.1002/alr.22330

Methods

- please adpat pictos framework

- All verbs in the text must be edited in the third person. For now we write '' We can do the research ... and changing it also in the past: a research has been carried out

Discuss the bias analysis and majors tools used.

Discussion

line 199, The inflammatory nasal mucosa therefore loses the function of mucociliary clearance, humidification and filter, the patient has nasal obstruction and does not respond over time even to standard medical therapy. please discuss and cite doi:10.1007/s00405-022-07267-0

Reviewer #2: This protocol is an interesting proposal of research concerning a widespread postoperative issue after Endoscopic Sinus Surgery for chronic rhinosinusitis.

However, there are some major limitations about this topic and this specific protocol.

First, TCM nasal irrigations are not clearly and accurately defined in a unique composition. This means that it can be difficult to compare a control group to an intervention group, because the intervention must be a precisely defined medication.

Second, possible disadvantages or complications related to the use of TCM irrigations are not mentioned (i.e. infections) but they are paramount when evaluating the indications to the interventions.

Further comments are pointed out below:

- Abstract, lines 24-25: do the Authors mean “after endoscopic sinus susrgery (ESS) for chronic rhinosinusitis (CRS)”? please correct if affirmative

- Introduction, line 65: I would consider to list every single drug category indicated in CRS or in alternative do not list them at all.

- Line 74: “has the characteristics of more minor side effects and easy operation” please revise the English form and the overall concept of the sentence

- Citation 8 and 9 have the same title but different Author; reference nr. 9 is not available in the international databases, the nr. 8 is a Chinese language article; please consider international and English language references. The same comment is related also to References nr. 10, 12, 13, 14, 15, 16 (not available in the international literature or non existent / incorrect DOI)

- Line 80: please replace “has” with “causes”

- Line 87: long-term use is not the point, please clarify the use of intranasal topical corticosteroids in the postoperative period

- In the section Inclusion criteria please state the chronological eligibility criteria of the studies

- Methods, Line 118: it is not clear what “simultaneously” means, consider explaining or removing it.

- Interventions section: control group should not include no treatment nor irrigations with glucocorticoids or antibiotics, but only saline irrigation in the postoperative time, while interventions should be represented only by saline irrigations with TCM.

7. PLOS authors have the option to publish the peer review history of their article (what does this mean?). If published, this will include your full peer review and any attached files.

Reviewer #1: No

Reviewer #2: No

---

## [Author Response · Author response to Decision Letter 0]

12 Jun 2022

Reviewer #1: The paper is interesting minor corrections could be improve the overall quality:

English editing should be performed.

I should change the titile in PLOS ONE

Efficacy and safety of traditional Chinese'' Effectiveness of traditiona..a systematic review..

Abstract

we used..not We will use Review Manager 5.3 software

Introduction

- Pediatric chronic rhinosinusitis (CRS) is a common disorder that carries significant morbidity. The diagnosis requires sinus symptoms that persist despite standard medical therapy greater than 3 months. Viral infections, allergies, and anatomic differences in children lead to chronic obstruction of the osteomeatal complex. The diagnosis is a conglomeration of multiple phenotypes and endotypes. As such, the diagnosis and management are complex. The emerging potential treatment options of CRS, including anti-immunoglobulin E, interleukin-5, and interleukin-4 receptor alpha subunit. please discuss and cite doi:10.1007/s11882-018-0792-8

- line 63, An interesting systematic review on single-nucleotide polymorphisms and risk-related chronic rhinosinusitis reporting the gene variation implicated in the pathogenesis of chronic inflammation and polyps. 12 papers with 9127 patients, of which 2739 CRS cases and 6388 controls were found. The major comorbidities reported related to chronic rhinosinusitis were atopy in 4555 (49.9%), asthma in 4594 (50.33%), Samter Triad in 448 (4.9%) and eosinophilia in 391 subjects (4.28%). please cite doi:10.1111/coa.13870

- line 67, Endoscopic treatment represents the best therapeutic option that the surgeon can offer for invasiveness and safety, allowing quick post-surgical recovery, less postoperative pain and fewer complications. please discuss and cite doi:10.1007/s00405-021-06724-6

- line 72, Nasal saline irrigation (NSI) plays an important role in the treatment of chronic rhinosinusitis (CRS). It is a beneficial low-risk treatment that serves an adjunctive function in the medical and surgical management of CRS. NSI is hypothesized to function by thinning mucous, improving mucociliary clearance, decreasing edema, and reducing antigen load in the nasal and sinus cavities. Although its use in CRS is nearly universal, significant variety exists with regard to delivery volume, delivery pressure, frequency of use, duration of use, composition, and hygiene recommendations. Evidence is limited regarding the most optimal methods of NSI delivery. In addition, use of NSI has recently come under increasing scrutiny due to potential associations with cases of primary amebic meningoencephalitis. An interestin review provided a clinical update summarizing use of NSI for treatment of CRS, including current recommendations for use, and data regarding overall efficacy, available delivery devices, solution composition, and hygiene. please discuss and cite doi:10.1002/alr.22330

Methods

- please adpat pictos framework

- All verbs in the text must be edited in the third person. For now we write '' We can do the research ... and changing it also in the past: a research has been carried out

Discuss the bias analysis and majors tools used.

Discussion

line 199, The inflammatory nasal mucosa therefore loses the function of mucociliary clearance, humidification and filter, the patient has nasal obstruction and does not respond over time even to standard medical therapy. please discuss and cite doi:10.1007/s00405-022-07267-0

Reviewer #2: This protocol is an interesting proposal of research concerning a widespread postoperative issue after Endoscopic Sinus Surgery for chronic rhinosinusitis.

However, there are some major limitations about this topic and this specific protocol.

First, TCM nasal irrigations are not clearly and accurately defined in a unique composition. This means that it can be difficult to compare a control group to an intervention group, because the intervention must be a precisely defined medication.

Second, possible disadvantages or complications related to the use of TCM irrigations are not mentioned (i.e. infections) but they are paramount when evaluating the indications to the interventions.

Further comments are pointed out below:

- Abstract, lines 24-25: do the Authors mean “after endoscopic sinus susrgery (ESS) for chronic rhinosinusitis (CRS)”? please correct if affirmative

- Introduction, line 65: I would consider to list every single drug category indicated in CRS or in alternative do not list them at all.

- Line 74: “has the characteristics of more minor side effects and easy operation” please revise the English form and the overall concept of the sentence

- Citation 8 and 9 have the same title but different Author; reference nr. 9 is not available in the international databases, the nr. 8 is a Chinese language article; please consider international and English language references. The same comment is related also to References nr. 10, 12, 13, 14, 15, 16 (not available in the international literature or non existent / incorrect DOI)

- Line 80: please replace “has” with “causes”

- Line 87: long-term use is not the point, please clarify the use of intranasal topical corticosteroids in the postoperative period

- In the section Inclusion criteria please state the chronological eligibility criteria of the studies

- Methods, Line 118: it is not clear what “simultaneously” means, consider explaining or removing it.

- Interventions section: control group should not include no treatment nor irrigations with glucocorticoids or antibiotics, but only saline irrigation in the postoperative time, while interventions should be represented only by saline irrigations with TCM.

---

## [Editor Report · Decision Letter 1]

19 Jul 2022

Efficacy and safety of traditional Chinese medicine nasal irrigation on chronic rhinosinusitis recovery after endoscopic sinus surgery：A protocol for a systematic review and meta-analysis

PONE-D-22-04435R1

Dear Dr. Zhang,

We’re pleased to inform you that your manuscript has been judged scientifically suitable for publication and will be formally accepted for publication once it meets all outstanding technical requirements.

Kind regards,

Giannicola Iannella, M.D

Academic Editor

PLOS ONE

Additional Editor Comments (optional):

the authors well improved the manuscript after reviewers comments.
---

## [Editor Report · Acceptance letter]

22 Jul 2022

PONE-D-22-04435R1 

*Efficacy and safety of traditional Chinese medicine nasal irrigation on chronic rhinosinusitis recovery after endoscopic sinus surgery： A protocol for a systematic review and meta-analysis*

Dear Dr. Zhang:

I'm pleased to inform you that your manuscript has been deemed suitable for publication in PLOS ONE. Congratulations! Your manuscript is now with our production department. 

Kind regards, 

on behalf of

Dr. Giannicola Iannella 

Academic Editor

PLOS ONE